**Article** https://doi.org/10.1038/s41467-024-51417-3

# The solvation shell probed by resonant intermolecular Coulombic decay

Rémi Dupuy [1,2] ✉, Tillmann Buttersack [1], Florian Trinter [1,3], Clemens Richter[1], Shirin Gholami[1], Olle Björneholm[4], Uwe Hergenhahn [1], Bernd Winter[1] & Hendrik Bluhm[1]

Molecules involved in solvation shells have properties differing from those of the bulk solvent, which can in turn affect reactivity. Among key properties of these molecules are their nature and electronic structure. Widely used tools to characterize this type of property are X-ray-based spectroscopies, which, however, usually lack the capability to selectively probe the solvation-shell molecules. A class of X-ray triggered "non-local" processes has the recognized potential to provide this selectivity. Intermolecular Coulombic decay (ICD) and related processes involve neighbouring molecules in the decay of the X-ray-excited target, and are thus naturally sensitive to its immediate environment. Applying electron spectroscopy to aqueous solutions, we explore the resonant flavours of ICD and demonstrate how it can inform on the first solvation shell of excited solvated cations. One particular ICD process turns out to be a potent marker of the formation of ion pairs. Another gives a direct access to the electron binding energies of the water molecules in the first solvation shell, a quantity previously elusive to direct measurements. The resonant nature of the processes makes them readily measurable, providing powerful new spectroscopic tools.

The formation of a solvation shell around ions and molecules dissolved in a liquid is the basis of many properties of solutions. Solute–solvent interactions, spanning from weak dipole-induced interactions to strong covalent bonding, affect both moieties, and thus, solvent molecules in a solvation shell may exhibit geometric and electronic structures, and dynamics, significantly different from other solvent molecules[1–3]. Consequently, reactivity may also be affected. Probing these differences at the molecular level using spectroscopic techniques is, however, challenging, as one needs a means to selectively probe the solvation-shell molecules, the signal of which is otherwise obscured by the much higher contribution from bulk solvent molecules. Vibrational spectroscopy and NMR spectroscopy are examples of commonly used tools[3]. In the field of liquid-phase electron spectroscopy, which has been evolving into a powerful tool of investigation in the last 20 years[4,5], so-called

"non-local relaxation" processes can be leveraged to achieve a specific probing of the solvation shell.

In liquids and other condensed-phase systems, excitation and ionization of inner-shell electrons by X-rays may lead to relaxation via non-local pathways alternative to the well-known local Auger and fluorescence decays. These non-local pathways involve the immediate environment of the excited or ionized species. Intermolecular Coulombic decay (ICD) is the prototypical example[6–8], but a host of variants, such as electron-transfer-mediated decay (ETMD)[9] or proton-transfer-mediated charge transfer (PTM-CS)[10], have been found. Cascades of these processes, mixed in with local Auger and fluorescence decay steps, may occur when deep core levels are ionized, leading to the emission of several electrons of various (often low) kinetic energies and the creation of several highly reactive ions[11,12]. In the last 20 years, the physics of these processes has been

[1]Fritz-Haber-Institut der Max-Planck-Gesellschaft, Faradayweg 4-6, 14195 Berlin, Germany. [2]Sorbonne Université, CNRS, Laboratoire de Chimie Physique - Matiere et Rayonnement, LCPMR, F-75005 Paris Cedex 05, France. [3]Institut für Kernphysik, Goethe-Universität Frankfurt, Max-von-Laue-Str. 1, 60438 Frankfurt am Main, Germany. [4]Department of Physics and Astronomy, Uppsala University, Box 516, SE-751 20 Uppsala, Sweden. ✉e-mail: remi.dupuy@sorbonne-universite.fr

studied in detail on model cluster systems, while a growing body of work is developing on liquids and a few other condensed-phase systems, as detailed in a recent exhaustive review[13]. Non-local processes have been recognized as an additional mechanism of radiation damage[14], and a great part of these works was motivated by that context.

It was nonetheless recognized early on that ICD-like processes could also serve as new spectroscopic tools. These processes are intrinsically sensitive to, and thus can inform on, the local geometric structure (number and distance) and electronic structure of the neighbours of the ionized or excited species. Some structural studies of clusters have been conducted[15,16] to demonstrate this. For instance, one of the most fundamental insights one can obtain concerns the chemical composition of this first coordination, i.e., solvation shell. In aqueous solutions, this can be extended to address ion pairing— whether a probed ion has a counter-ion in its immediate vicinity. Early attempts looking at non-resonant ICD and ETMD did not find evidence of ion pairing in concentrated potassium halide[17] or lithium chloride[9] solutions. The first evidence was found in the comparison of Li+ ETMD signals from lithium-chloride and lithium-acetate solutions[18], with an additional spectral signature being identified as an ion-pairing signature in the lithium-acetate case. These studies were impeded by the low cross-section of the processes under study and their occurrence at low kinetic energies, leading to poor signal-to-noise ratios. In liquids, a host of features related to solvents and solutes can occur across the spectral range and may overlap weak signals in the electron spectrum. In addition, a large inelastic-scattering background is always present and particularly hampers low-kinetic-energy measurements[19]. The use of resonant ICD partially overcomes these issues due to a much stronger signal, thus potentially enabling a more robust route to investigate ion pairing, as we will demonstrate in the present work.

Furthermore, the sensitivity of ICD processes to the neighbours' electronic structure uniquely offers direct access to the electron binding energies of molecules in the first solvation shell. Valence-band energetics are crucial quantities determining reactivity in solution[2] and can be expected to be different for solvent molecules involved in hydration shells compared to bulk solvent molecules[20]. An experimental quantitative distinction between these two contributions has not been possible until now, because in liquid-jet photoelectron spectroscopy, photoelectrons from all molecules are probed simultaneously, with the

hydrating-molecules' contribution being the smaller one. A few studies investigated highly concentrated solutions[21,22], where almost all water molecules are solvation-shell molecules; however, in such cases, the solvent molecules are shared between cations and anions, and the hydrogen-bond network is significantly disrupted.

A recent study[20] has shown that (non-resonant) ICD of solvated ions, in this case Mg2+ and Al3+, has the potential to measure the valence electronic structure of the hydrating molecules. However, the quantitative determination of electron binding energies was hampered by additional contributions that could not be accurately quantified. Indeed, in non-resonant ICD, the final state consists of one valence hole localized on the solute and another one on the neighbouring water molecule, and the Coulomb repulsion between the ions adds to the energetics of the process. With resonant ICD, this problem is shown here to be bypassed, enabling direct measurement of the electron binding energies of hydrating water molecules, as demonstrated here for the case of solvated Ca2+, one of the most important divalent ions in biological systems[23].

Resonant ICD can manifest itself as two different variants, in complete analogy with resonant Auger decay, where participator and spectator processes are distinguished depending on whether the excited electron is involved in the decay process or not. Participator and spectator ICD are shown schematically in Fig. 1. Although these terms were used in early theoretical discussions of the process[24], the bulk of ICD literature has used a different terminology. In clusters, the spectator process was studied early on and termed simply "resonant ICD" or RICD[25–29]. The participator process was discussed theoretically under the name "two-centre resonant photoionization" (2CPI)[30], with a few recent experimental demonstrations, also in model rare-gas cluster systems[31–34]. Resonant ICD has been overall little explored in liquids, with the notable exception of the first liquid-phase ICD work[35].

In contrast to the previous examples, in our investigation both processes are detected and are in competition with local resonant Auger decay. We explored the case of resonant 2p to 3d excitation of Ca2+ ions solvated in water. We use the complementary aspects of the two resonant ICD processes as a means to probe the first solvation shell of Ca2+. Spectator ICD provides a powerful probe of ion pairing, i.e., the composition of the first shell, mainly because the strong resonant signal is much more reliably normalized and compared across different solutions than for previously measured weak

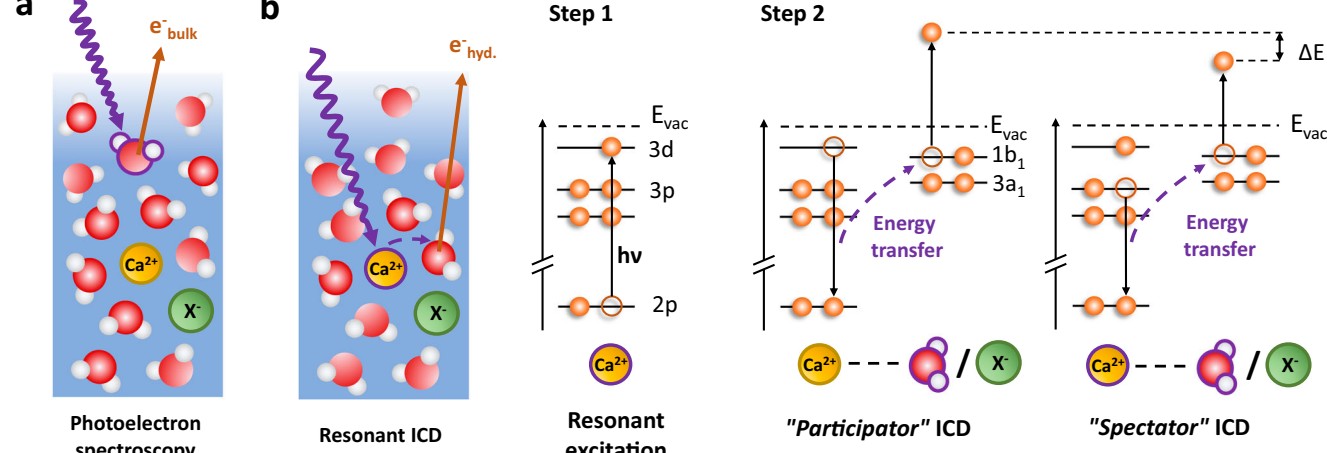

**Fig. 1 | Schematic diagram of resonant ICD processes in solvated calcium.**
**a** Regular photoelectron spectroscopy probes electrons emitted from any molecule in the solution. The purple wave arrow represents an incoming photon and the orange arrow represents an outgoing electron. **b** In resonant ICD, electrons are emitted specifically from the solvation shell of the excited Ca2+ ion. The first step is the resonant excitation of Ca 2p electrons to empty 3d orbitals. In the second step,

ICD relaxation occurs by energy transfer and ionization of a neighbouring water molecule (or counter-ion X−). One can distinguish between the participator case (the excited electron fills the 2p hole) and the spectator case (another calcium valence electron fills the 2p hole). The emitted ICD electron has a different kinetic energy for the two processes.

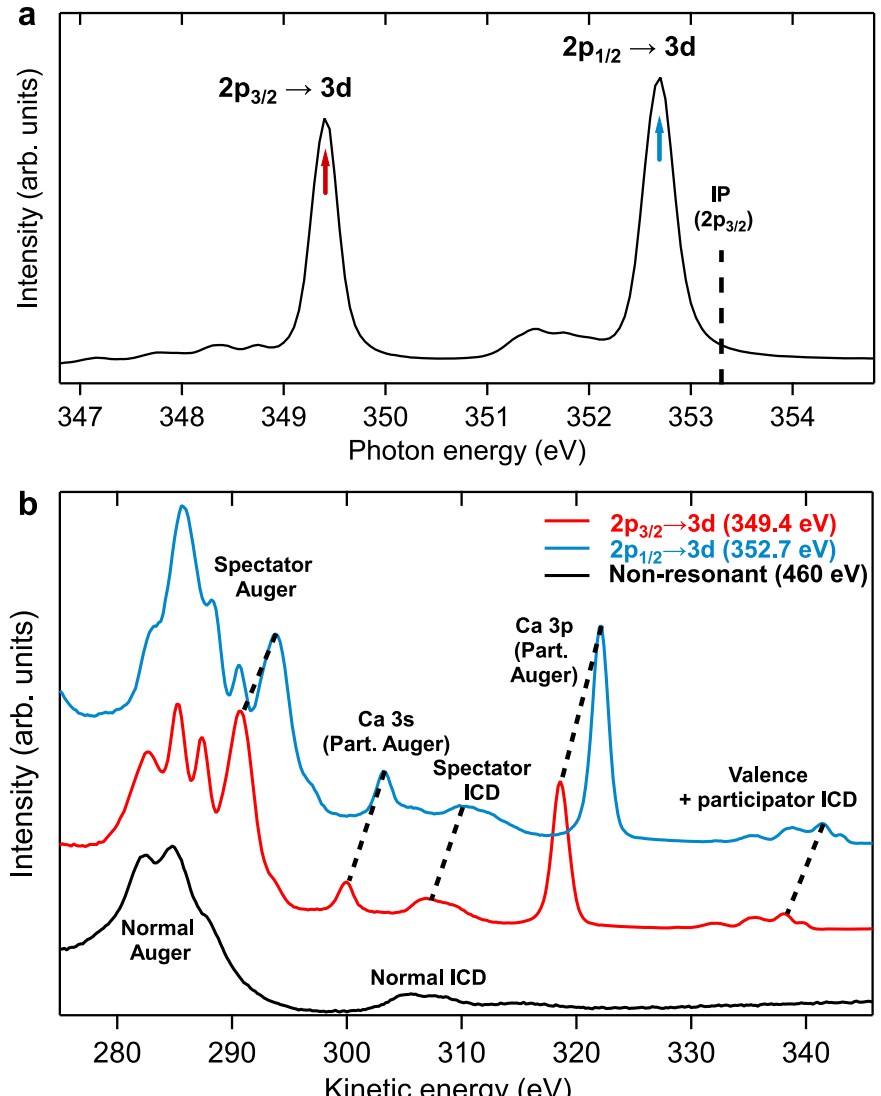

**Fig. 2 | Resonant absorption and decay of solvated $Ca^{2+}$. a** X-ray absorption spectrum (XAS) at the Ca 2p edge of a 1.5 M $CaCl_2$ solution. The spectrum is recorded in partial-electron-yield mode by integrating electron counts over the 270–350 eV range. IP ionization potential. **b** Resonant Ca 2p Auger and ICD spectra of the same solution. The two resonant spectra (red and blue traces) correspond to the excitation energies of the two main peaks of the panel (**a**) (red and blue arrows). The non-resonant Auger/ICD spectrum at 460 eV is also shown (black trace) for comparison. The most important attributions are made in the figure, with dashed black lines following the resonant features as they shift with photon energy. The spectra have been scaled and offset on the intensity scale to improve visibility.

non-resonant processes[9,18]. Participator ICD can reveal the electronic structure of the first-shell water molecules because the final state here is a one-hole state, akin to direct photoemission, and thus is free from the previously mentioned unknown Coulomb repulsion terms of normal ICD. Both processes constitute particularly interesting additions to the liquid-phase photoemission toolbox.

## Results and discussion
### Resonant Auger and ICD spectra
We aim to study the first solvation shell of the $Ca^{2+}$ ion in water. Electronic decay processes in a 1.5 M $CaCl_2$ aqueous solution are studied by coupling liquid-microjet electron spectroscopy with synchrotron radiation as a tuneable source of soft X-rays[36], as detailed in the "Methods" section. The initial step is the resonant excitation of Ca 2p electrons to unoccupied Ca 3d orbitals. The corresponding absorption spectrum at the Ca 2p edge is shown in Fig. 2a. The several features of the spectrum are discussed in the literature[37,38] and arise from the reduced symmetry of the hydrated ion compared to the isolated one. Only the two most intense peaks will be probed here.

In Fig. 2b, electron spectra obtained at these two resonance peaks, in the region where Auger decay and ICD manifest, are shown. A non-resonant Auger and ICD spectrum above the 2p threshold, in good agreement with spectra already reported in the literature[39], is also displayed for comparison. In the latter, normal ICD appears at kinetic energy ~20 eV higher than normal Auger features, roughly reflecting the energy difference between the binding energies of the Ca 3p shell and the water valence shell. While resonant spectra have not yet been reported for solvated $CaCl_2$, a comparison can be made with reported 2p resonant spectra of atomic Ca[40] and solid $CaCl_2$ films[41]. In the resonant spectra in Fig. 2, the features in the 280–295 eV range can be assigned to spectator Auger (at similar but slightly blue-shifted energies compared to the normal Auger). Similarly, the feature around 305–310 eV, at slightly blue-shifted energies compared to the normal ICD, is assigned to spectator ICD. Features are also observed in the resonant spectra of atomic Ca and solid $CaCl_2$ in this energy range; there, they correspond to Auger processes in which a Ca 3p electron and a valence electron of the system are involved. This is quite similar to what we observe here, except that the valence electron must be

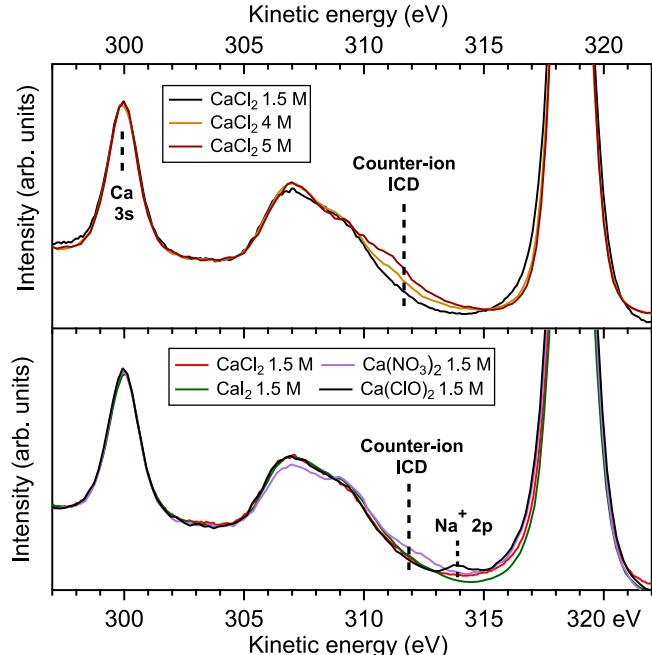

**Fig. 3 | Observation of a signature for ion pairing in the spectator ICD spectra at 349.4 eV (main $2p_{3/2} \rightarrow 3d$ line, see Fig. 2) for several calcium salt solutions.** In the top panel, $CaCl_2$ solutions of different concentrations are explored. Concentrations lower than 1.5 M yield spectra similar to the 1.5 M one (see SI). In the bottom panel, 1.5 M $Ca^{2+}$ solutions with different counter-ions are displayed. Details on the solution preparation [explaining, for instance, the $Na^+$ contamination in the $Ca(ClO)_2$ solution] are available in the "Methods" section.

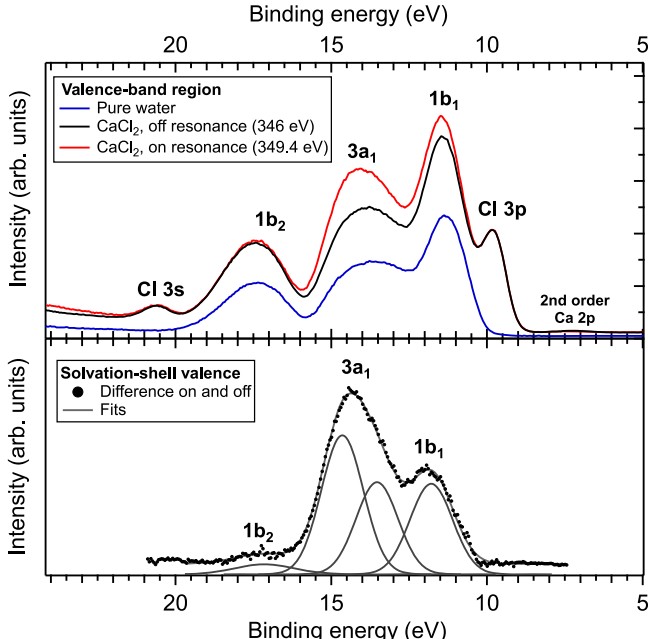

**Fig. 4 | Determination of the valence-band spectrum of water molecules in the solvation shell of $Ca^{2+}$.** In the top panel, the valence-band region of the electron spectra off-resonance (at 346 eV) and on the main $2p_{3/2} \rightarrow 3d$ resonance (at 349.4 eV) for the $CaCl_2$ 1.5 M solution are displayed. The valence band spectrum of pure water is also displayed for comparison. The different orbitals are attributed in the figure. The differences are attributed to the resonant participator ICD. The two spectra have been normalized so that the Cl $3p$ and $3s$ peaks overlap, as Cl⁻ is not involved in participator ICD. In the bottom panel, the difference between the two spectra, corresponding to the participator ICD contribution only, is displayed along with a fit of the different peaks. This spectrum reflects the valence band of the $Ca^{2+}$ solvation-shell water molecules. A contamination by Ca $2p$ photoionization with second-order light is notable in the 346 eV spectrum.

borrowed from a neighbouring water molecule without any chemical bond to calcium, which is characteristic of ICD.

Participator ICD (Fig. 1) should manifest itself as an on-resonance enhancement of the valence-band features of water. Indeed, as long as there is no significant nuclear dynamics in the intermediate state, participator ICD is energetically equivalent to direct ionization of the valence-band features, much like participator Auger is energetically equivalent to direct ionization of the outermost orbitals of Ca, and is thus identified by an on-resonance enhancement of the Ca $3s$ and $3p$ photoelectron lines in Fig. 2. The on-resonance enhancement of the water valence band is not obvious in Fig. 2; it will become more evident in another figure shown below. From the integration of the peak areas of the different features in the spectra, one can estimate that spectator ICD represents ~4% of all decay processes, while participator ICD represents <0.5%.

**Spectator ICD: probing the nature of the solvation shell**

In order to investigate the sensitivity of resonant ICD to ion pairing, we systematically varied the anion with which calcium is paired in solution (Cl⁻, I⁻, ClO⁻, or $NO_3^-$), as well as, for a $CaCl_2$ solution, the solution concentration. The results of these studies are shown in Fig. 3. We focus here on the spectator ICD feature. All spectra are normalized to the Ca $3s$ peak. While the main group of broad features in the 305–310 eV kinetic energy (KE) range corresponds to spectator ICD with hydration-shell water molecules, in several cases, one can observe the presence of an additional high-kinetic-energy shoulder (~312 eV KE) in the spectrum. This additional feature can be interpreted as ICD with the counter-anion. Indeed, most anions in solution have the highest occupied molecular orbital (HOMO), which is lower in binding energy than the valence band of water[42]. The water HOMO, the $1b_1$, is found at 11.33 eV binding energy[43], while, for instance, Cl⁻ $3p$ is measured at 9.8 eV binding energy in the 1.5 M $CaCl_2$ solution. A lower HOMO binding energy translates into a higher KE of the ICD electron,

as can be understood from the energy balance of the process: $E_K = h\nu - E_B^{partner} - E_{exc} - E_R$, where $E_K$ is the kinetic energy of the ICD electron, $E_B^{partner}$ the binding energy of the orbital of the ICD partner (water or anion), $E_{exc}$ is the energy required to excite a $3p$ electron to $3d$ in $Ca^{2+}$ and $E_R$ is an (unknown) repulsive term. The ICD spectrum thus reflects the valence electronic structure of the ICD partners.

The occurrence of ICD with the counter-anion as a partner implies that this anion is present within the first coordination shell of the cation, indicating contact ion pairing. For instance, in the top panel of Fig. 3, we observe the appearance of the ICD feature with Cl⁻ for a 4 M concentration, and it is even more pronounced at 5 M. In the literature, X-ray diffraction[44] and EXAFS[45] studies indeed found an onset of contact ion pairing around 4 M, in contradiction with a neutron-diffraction study[46] which found no evidence of ion pairing up to the solubility limit of 6.4 M. Our results are in agreement with the first two studies, supporting their conclusion of an onset of ion pairing around 4 M for $CaCl_2$ solutions.

In the bottom panel, we kept the $Ca^{2+}$ concentration constant at 1.5 M but varied the counter-ion. At this concentration, only for $NO_3^-$ we observe a trace of counter-ion ICD and thus of ion pairing. This is again in agreement with the literature, where ion pairs between calcium and nitrate were reported[47]. No data could be found on pairing between calcium and iodide or hypochlorite ions, and our results suggest that no contact ion pairing occurs at 1.5 M.

While all types of ICD and other related processes are, in principle, sensitive to ion pairing, spectator ICD is much easier to detect due to its resonant nature. In the present case, it is possible to obtain information from the normal ICD spectra, as we show in the SI, but at the expense of several difficulties, which are also listed in the SI.

**Table 1 | Peak positions and widths of the valence orbitals of water**

| | Peak position (eV) | | | | FWHM (eV) | | | |
|---|---|---|---|---|---|---|---|---|
| | $1b_1$ | $3a_1$L | $3a_1$H | $1b_2$ | $1b_1$ | $3a_1$L | $3a_1$H | $1b_2$ |
| Pure water (literature) | 11.33 | 12.95 | 14.38 | 17.49 | 1.45 | 1.62 | 1.62 | 2.40 |
| Pure water (this work) | 11.27 | 12.97 | 14.37 | 17.43 | 1.42 | 1.62 | 1.62 | 2.41 |
| CaCl$_2$ 1.5 M | 11.41 | 13.27 | 14.57 | 17.47 | 1.64 | 1.53 | 1.53 | 2.44 |
| Ca$^{2+}$ hydration shell | 11.85 | 13.57 | 14.68 | | 1.47 | 1.60 | 1.60 | |

Values are given for pure water (literature[43] and this work) and 1.5 M CaCl$_2$, as measured by regular (off-resonant) photoelectron spectroscopy, and for the calcium hydration shell, measured via participator ICD. For the latter, the $1b_2$ peak is omitted, as it is too weak for a reliable determination of the peak parameters. The uncertainty associated with these values is discussed in the SI. The uncertainty of the binding energies (absolute peak positions) associated with the uncertainties of the kinetic and photon energies are ±0.05 eV. Typical uncertainties in determining peak positions and FWHM by the fitting method are ±0.01 and ±0.05 eV, respectively, but can be as high as ±0.2 eV for the last row of the table (see SI).

## Participator ICD: probing the electronic structure of the solvation shell

We now turn our attention to the second ICD process: participator ICD. As mentioned above, this process should manifest itself in the valence-band region as an enhancement of the photoelectron lines of the ICD partners, because the final state is the same as for direct photoemission from the valence orbitals. In Fig. 4, we show the valence-band region of water, where the three outer-valence orbitals of water, $1b_1$, $3a_1$, and $1b_2$, as well as the Cl $3s$ and $3p$ orbitals, are observed. In condensed-phase water, the $3a_1$ peak is split into two components, $3a_1$L and $3a_1$H, due to intermolecular interactions[48–50]. We display this valence region for on-resonance excitation at 349.4 eV and below the Ca $2p$ resonance region at 346 eV. The spectra are intensity-normalized at the Cl$^-$ peaks. In principle, as outlined in Fig. 1, participator ICD with Cl$^-$ counter-ions is also possible. However, we discussed above that no ion pairs are formed in 1.5 M CaCl$_2$ solutions, and therefore, no Cl$^-$ ICD is detected in the spectator ICD feature. We can, therefore, also assume the absence of Cl$^-$ ICD in the valence/participator ICD signal, and normalize these spectra to the Cl$^-$ peak. One can, in this way, clearly observe an on-resonance excess of intensity on the $1b_1$ and $3a_1$ peaks, which cannot be explained by the very small variations of relative photoionization cross sections and, therefore, is the signature of participator ICD.

The bottom panel of Fig. 4 shows the difference between the on- and off-resonance spectra, thus displaying only the contribution of participator ICD. In this spectrum, the different orbitals of water are easily distinguished, which is not the case in the spectator ICD spectrum and neither was possible in previous non-resonant (normal) ICD studies[20]. With spectator or normal ICD, the final state also features the excited or ionized neighbouring cation, and the electronic structure of the ICD partner is not revealed as clearly, although it is still underlying the ICD spectrum.

In the participator ICD spectrum, we observe relative peak intensities quite different from the direct photoionization spectrum, as well as different relative and absolute peak positions. In addition, fitting the spectrum also reveals some differences in the peak widths, although this parameter was found to be sensitive to the subtraction procedure. The sensitivity of the different parameters to the subtraction method is discussed in the SI. These differences can all be rationalized by qualitative arguments, while comparison to theoretical calculations will be the subject of future work. The intensity of the different features is related to the factors that influence the efficiency of the ICD process (and not just the ionization cross sections, as in regular photoionization). On the other hand, their positions and widths directly reflect the electronic structure of the hydration-shell molecules, something that could not be accessed experimentally

before. This point was already highlighted in the calculated results of ref. 20. We will discuss both aspects below.

From the relative intensities, we can deduce that participator ICD is more efficient for the $3a_1$ orbital than for the $1b_1$ orbital, and it is very weak for the $1b_2$ orbital. A propensity of non-local decay from the $3a_1$ orbital has also been seen in previous works on non-resonant ICD[20] and ETMD (electron-transfer-mediated decay)[9] in solution. As reported here, the ICD efficiency is sensitive to intermolecular distance and, in a regime of close proximity as may be expected for solvation shells (in contrast to, e.g., weakly bound van der Waals dimers), electronic densities overlap. The $1b_2$ orbital is a bonding orbital with its electronic density mostly localized on the O–H bonds of the water molecule, and thus it is expected that ICD is not very efficient for this channel compared to the mostly non-bonding $3a_1$ and $1b_1$ orbitals, with considerable electronic density on the oxygen lone pairs. Indeed, in the solvation shell of a cation like Ca$^{2+}$, water molecules are oriented with the water dipole facing away from the cation, and thus the oxygen lone pairs towards it. The relative efficiency of ICD of the $3a_1$ and $1b_1$ orbitals should depend on their respective orbital overlap with the Ca$^{2+}$ orbitals. Since these two orbitals are orthogonal, we make the hypothesis that this relative efficiency is directly governed by the distribution of dihedral angles of the water molecules in the solvation shell. The dihedral angle is the angle formed by the Ca–O segment and the plane containing the water molecule. It is the second parameter that, together with the Ca–O distance, fully determines the geometric structure of the hydration shell. An angle of 0° will favour $3a_1$ orbital overlap, while an angle of 90° will favour $1b_1$ orbital overlap. The average angle for solvated Ca$^{2+}$ as deduced from neutron diffraction, is around 34°[51]. This argument is in the spirit of a recent work on solvated Cu$^{2+}$, which showed the sensitivity of the Cu $2p$ photoelectron satellites to the dihedral angle[52]. This is an example of the fact that the ICD spectrum also contains information on the geometric structure of the hydration shell.

As mentioned above, the peak positions and widths of the participator ICD spectrum reflect the electronic structure of the Ca$^{2+}$ solvation-shell molecules. The decisive question is to what extent this electronic structure differs from that of the average water molecule in the bulk solution, as probed by normal photoelectron spectroscopy. In Table 1, we collected the positions and widths for the different components of the water valence band, measured on pure water and 1.5 M CaCl$_2$ (off-resonance) by photoelectron spectroscopy, and those deduced from the participator ICD spectrum. Absolute positions were determined from the cut-off method outlined in ref. 43 (see also the "Methods" section). Our measurement for pure water is in good agreement with the literature values[42,43].

The third row of the table corresponds to normal valence-band photoemission measurements of the 1.5 M CaCl$_2$ solution, i.e., the average water electronic structure in that solution. Only modest changes occur compared to pure water: peaks are blue-shifted on the order of 200 meV, the $1b_1$ peak is broadened, and the $3a_1$ peak splitting is reduced. This is in good agreement with previous works on the effect of electrolytes on the electronic structure of water, studied for NaI solutions[21,53] over a wide range of concentrations (0.5–8 M).

Significant differences are observed in the case of the participator ICD spectrum (fourth row in Table 1). The peak widths are close to the value for pure water within the precision with which they are constrained (see SI). The peak positions, on the other hand, clearly indicate a large blue-shift of about 500 meV of the $1b_1$ peak and the $3a_1$L peak, while the $3a_1$H peak is only slightly shifted, thus making the $3a_1$ split much smaller (1.1 eV instead of 1.4 eV in pure water). A blue-shift of this magnitude agrees well with the theoretical predictions made for solvated Mg$^{2+}$ in ref. 20 and, as pointed out in that reference, is the expected behaviour for cation hydration. Interpretation of the 1:0.5 intensity ratio of the $3a_1$H to $3a_1$L components will require more theoretical input, as in that case, electronic structure and ICD-efficiency

effects are intermingled, but it also indicates significant changes in the hydration shell. We can, therefore, conclude that the electronic structure of the hydration-shell water molecules is markedly different from that of the bulk water molecules. These changes can be directly accessed here, which was not possible before.

## Summary and conclusions

We have provided evidence for resonant ICD after core-level excitation of a solvated cation, $Ca^{2+}$, using liquid-jet photoemission spectroscopy. Resonant ICD has been little explored in liquid systems so far. Two distinct resonant ICD processes are identified here, which we term "spectator" and "participator" ICD, in full analogy to the corresponding resonant Auger processes. Previously, the participator process was only observed in model rare-gas cluster systems and in cases where it is the only expected electron-emission decay pathway, in competition with fluorescence[32,34]. In contrast, we observe here both spectator and participator processes, and both are in competition with the local resonant Auger decay pathways. Spectator ICD represents 4% of the total relaxation pathways, and participator ICD <0.5%.

Aside from this step forward in the fundamental understanding of ICD in liquids, we showed how these processes constitute valuable additions to the liquid-phase photoemission toolbox, providing probes of the solvation shell of the core-excited solutes. While all ICD variants contain, in principle, the same information about the nature, geometry, and electronic structure of the solvation shell, this information is easier to extract from certain processes. Signatures of the presence or absence of ion pairing between $Ca^{2+}$ and its anion partners are provided by spectator ICD. The resonant character of the process makes this signature easy to measure and unambiguous, providing a useful alternative to normal ICD and other ICD-like processes used thus far for tracking ion pairing[18]. Although participator ICD is also a resonant process, it is weaker and necessarily has to be detected on top of the regular valence photoemission features, making it less suitable for this purpose.

On the other hand, participator ICD provides the even more appealing and novel possibility of directly probing the valence electronic structure of the water molecules in the solvation shell, which we show in the case of $Ca^{2+}$ to differ from that of the average water electronic structure as probed by direct photoelectron spectroscopy. Important quantities, such as the electron binding energies of these solvation-shell molecules, now become directly measurable. Inferences can also be made on the geometric structure of the water molecules (dihedral angle) from the relative involvement of different water orbitals in the participator ICD process.

## Methods
### Experimental details
The measurements were performed using the EASI instrument[36] for liquid-jet photoemission studies, installed at the P04 beamline[54] of the PETRA III synchrotron. Briefly, the experimental setup is equipped with a liquid microjet and a hemispherical electron analyser (HEA) to perform liquid-phase photoemission. The liquid microjet was typically operated with a ~35 μm diameter quartz nozzle and a delivery flow rate of 1 mL/min. The HEA was operated at pass energies of either 50 or 100 eV, with entrance-slit settings yielding an analyser resolution of respectively ~100 or ~200 meV. The jet propagates in the horizontal direction at 90° from the photon beam. The HEA is placed outside the dipole plane, at 130° from the beam propagation direction in a back-scattering geometry.

The beamline was operated with a monochromator exit slit of 150 μm, delivering a beam with a photon bandwidth of 140 meV at 400 eV and a spot size of ~50 μm × 200 μm (vertical × horizontal). The spot thus overlaps optimally with the microjet. The photon flux is up to $3 \times 10^{12}$ photons/s at this photon energy and exit slit and can be tuned down with thin foil filters or baffles between the undulator and the first beamline mirror when necessary. The photon energy was calibrated at several values, including near 400 eV, using well-known gas-phase absorption resonances (CO, $N_2$, $SF_6$, and Ne).

Measured electron kinetic energies were corrected using the calibration procedure described recently in ref. 43. The measurements were performed with a bias voltage of −50 V applied on the liquid microjet and low-energy electron cut-off spectra were measured to determine the absolute zero of kinetic energy with respect to the local vacuum level of the microjet. This procedure additionally removes almost entirely gas-phase water contributions to the valence-band spectra.

### Solution preparation
$CaCl_2$ (Sigma-Aldrich, dihydrate ACS reagent 99%) and $CaI_2$ (Sigma-Aldrich, hydrate 98%) commercial salts were dissolved in Milli-Q water to obtain 1.5 M solutions and also 4 and 5 M solutions of $CaCl_2$. $Ca(ClO)_2$ commercial salts are only pure to 70% and thus contain significant amounts of impurities, mostly $CaCl_2$ and $Ca(OH)_2$. The commercial product (Sigma-Aldrich) was nonetheless used; non-soluble impurities [mostly $Ca(OH)_2$ in excess of its maximum solubility] were filtered out. Cl 2$p$ photoelectron spectra sign the presence of $Cl^-$ in a ratio of 1:5 with $ClO^-$ in the solution. The presence of a visible Na 2$p$ peak also signals non-negligible amounts of $Na^+$. The amount of dissolved commercial salt was calculated assuming pure $Ca(ClO)_2$ to yield a solution with approximately 1.5 M concentration of $Ca^{2+}$, but for the reasons outlined here, the actual concentrations of $Ca^{2+}$ and $ClO^-$ are uncertain. This does not affect the main conclusions of the study. $Ca(NO_3)_2$ was prepared by adding $Ca(OH)_2$ (Sigma-Aldrich, ACS reagent 95%) to a 3 M solution of nitric acid $HNO_3$ [prepared from dilution of a commercial 65 wt% solution (Sigma-Aldrich)] until pH neutralization (pH > 3 in this case). All solutions were filtered and degassed with an ultrasonic bath before injection through the microjet.

## Data availability
Data relevant to this study is available at https://doi.org/10.5281/zenodo.12807453. Numeric representations of the traces in the figures and raw data underlying the presented results are provided.

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

## Acknowledgements

R.D. acknowledges support from the Alexander von Humboldt Foundation through a Postdoctoral Fellowship. F.T. acknowledges funding by the Deutsche Forschungsgemeinschaft (DFG, German Research Foundation)—Project 509471550, Emmy Noether Programme. F.T. and B.W. acknowledge support by the MaxWater initiative of the Max-Planck-Gesellschaft. B.W. acknowledges funding from the European Research Council (ERC) under the European Union's Horizon 2020 research and innovation programme under grant agreement

No. 883759— AQUACHIRAL. O.B. acknowledges support from the Swedish Research Council via the grant 2023-04346. We acknowledge DESY (Hamburg, Germany), a member of the Helmholtz Association, HGF, for the provision of experimental facilities at PETRA III under proposals I-20211422, I-20220291, and I-20221212. We thank M. Hoesch and his team for assistance in using the P04 beamline. We acknowledge J. Buck, F. Diekmann, M. Kalläne, and S. Rohlf from Christian-Albrechts-Universität zu Kiel for providing us with software to scan the photon energy via our SCIENTA spectrometer software. We thank S. Thürmer for his Igor analysis procedures.

## Author contributions

R.D., T.B., F.T., C.R., S.G. and H.B. participated in the measurement campaigns. F.T., U.H. and B.W. designed, constructed, and provided the experimental setup. R.D., T.B., C.R., O.B. and H.B. conceived the experimental plans. R.D. analysed the data and wrote the manuscript with input from all authors. All authors discussed the results and commented on the manuscript.

## Funding

## Competing interests

The authors declare no competing interests.
