## [Peer Review File · Nature Communications]

The solvation shell probed by resonant intermolecular
Coulombic decayREVIEWER COMMENTS

Reviewer #1 (Remarks to the Author):

This paper shows how resonant intermolecular Coulomb decay (ICD) in liquid microjets has the potential to yield information about the electronic and geometric structure of the first solvation shell of ions and molecules in liquids. This is an important topic because the 1st solvation shell underpins many properties of, and processes in, solutions. It is therefore of interest to a broad chemistry audience.

The authors present a series of careful measurements of resonant ICD in Ca^{2+} solutions and describe clearly how the participator and spectator ICD processes have the potential to yield electronic and structural information, respectively, about the first solvation shell.

The paper is very well written so is accessible to a general chemistry audience. However, there are some points that could be clarified for it to be understood without recourse to the X-ray photoelectron spectroscopy literature.

For example, Fig. 1 is very valuable in explaining the processes involved in this work. It would be even more useful if the caption could explain that $\text{H}_2\text{O} / \text{X}^-$ in the second figure of Step 2 means that energy can apparently be transferred to either water or X^- in the spectator ICD, but not in the participator ICD. This was only stated explicitly on p5 of the manuscript.

It is also important that the authors explain why spectator ICD is a better probe of ion-pairing than participator ICD. This may be obvious to them, but to a non-expert in X-ray PES, it is not entirely obvious, and for this work to be published in a more general chemistry journal it would be nice to have it explained in the manuscript.

The discussion of the results is very clear. However, there are a couple of small things that should be addressed.

In the discussion of the effect of concentration (Fig. 3), the authors suggest that there is no ICD from the counter-ion in the 1.5 M spectrum, but it is not obvious from the data presented that this is necessarily the case. Was a spectrum recorded at weaker concentrations that had the same profile as the 1.5 M spectrum? If so, could this be reported in the text or the figure caption? If not, perhaps it would be better to state that there is less ICD from the counter-ion at 1.5 M rather than none.

Since the point is made about the peak positions and widths being able to reveal information about the electronic and geometric structure of the solvent shell, it would have been useful to have some quantitative idea of how reliable the widths are that are presented in Table 1. The authors note twice in the manuscript how they are very sensitive to the subtraction process—so some more reassurance about this would be valuable. Moreover, if the widths are very sensitive to the subtraction process, aren't the peak positions and relative intensities of the various peaks also sensitive to the subtraction process? Perhaps the sensitivity study of all these parameters, which has presumably been undertaken already, should be presented as Supplementary Information to support the arguments in the manuscript.

In Table 1, it is stated that the error on the measurements is 0.03 eV; however, the water 3a1 peak positions from the measurements reported in this paper (row 2) are not within the error limits of the literature values (row 1). Perhaps the authors could comment on this or revise their error bars.

Finally, in the Methods, the authors state that most measurements are made with a bias voltage of 50 V. What is meant by 'most'? And have the authors considered whether the bias voltage of 50 V has an impact on the alignment of the water molecules and counter-ions around the Ca^{2+} ions being studied?

Reviewer #2 (Remarks to the Author):

This manuscript 'The solvation shell probed by resonant intermolecular Coulombic decay' by Dupuy reports an experimental study on the first solvation shell of Ca^{2+} solvated in CaCl_2 aqueous solution using liquid-jet photoemission spectroscopy. To selectively probe the solvation shell of water molecules surrounding the Ca^{2+} divalent ion, the authors study two variants of resonant ICD (RICD) following core level $2p \rightarrow 3d$ resonant excitation of Ca^{2+} in the aqueous solution. They claim that "spectator RICD" informs about the chemical composition of the first solvation shell (also the signature of ion pairing between Ca^{2+} and its anion partner) and that "participator RICD" reveals the electronic structure of this solvation shell. So far, these two variants of RICD have only been identified theoretically in one system. The experimental observation of these RICD processes in a relevant system (liquid water) and the demonstration of their benefits for gaining information about solvation shells in liquids is valuable. The manuscript is well written, and the figures are clear. However, before we can consider recommending this manuscript for publication the following serious concerns must be dispelled.

Ca atoms and CaCl_2 have been studied by x-ray photoemission spectroscopy for a long time, see e.g. PRA 49 3685 (1994), PRB 47 11736 (1993).

1. The on-resonance electron spectra of Ca atoms (Fig. 7 of PRA 49 3685 (1994)) and CaCl_2 films (Fig. 5 in PRB 47 11736 (1993)) contain a feature between the Ca 3s and 3p photolines that resembles the feature assigned to "Normal ICD" or "Spectator ICD" in the current manuscript. What's the basis for the authors' claim that the feature they observe in the kinetic energy range 303-315 eV (Fig. 3) is due to ICD?
2. This same feature does not appear to be enhanced on resonance compared to the non-resonant case (Fig. 2 b, "normal ICD"). Only the edge of that feature around 312 eV kinetic energy is enhanced for high CaCl_2 concentrations and for $\text{Ca}(\text{NO}_3)_2$ solution. What's the basis of the authors' claim that this signal is due to "spectator RICD"? Is this concentration-dependent signal around 312 eV present off resonance (normal ICD) as well? If the authors want to uphold the claim that RICD is advantageous they have to clearly show/discuss this.
3. The authors should rationalize how the normal ICD / RICD signal of the Cl^- counterion can reach a level of $\sim 50\%$ of the H_2O ICD signal given the overall small concentration of Cl^- ions relative to water molecules (no depletion of the H_2O ICD signal is seen). Is ICD to Cl^- so much more efficient? Are there more than 2 Cl^- ions in the solvation shell around Ca^{2+} ? Is there any other signature of Counter-ion ICD that leads to detachment of the bound electron that forms the Cl^- anion?
4. The authors mention that electron inelastic scattering tends to be an issue when probing bulk liquid samples; What is the role of inelastic scattering of the participator RICD electrons, in particular in the discussion of the peak positions for participator ICD (Fig. 4)?
5. It is surprising to see a strong resonant enhancement of participator RICD of H_2O but no resonant enhancement of Cl^- (Fig. 4). Showing data for higher concentrations where participator RICD of Cl^- is also resonantly enhanced would make the case more convincing.
6. Overall, the discussion and conclusions obtained from these measurements remain purely qualitative. In the case of spectator ICD the only conclusion is that one previous observation is supported vs. another. The manuscript would certainly be more impactful if more quantitative information could be extracted.

Reviewer #3 (Remarks to the Author):

Listed Report:

This manuscript 'The solvation shell probed by resonant intermolecular Coulombic decay' by Dupuy reports an experimental study on the first solvation shell of Ca^{2+} solvated in CaCl_2 aqueous solution using liquid-jet photoemission spectroscopy. To selectively probe the solvation shell of water molecules surrounding the Ca^{2+} divalent ion, the authors study two variants of resonant ICD (RICD) following core level $2p \rightarrow 3d$ resonant excitation of Ca^{2+} in the aqueous solution. They claim that "spectator RICD" informs about the chemical composition of the first solvation shell (also the signature of ion pairing between Ca^{2+} and its anion partner) and that "participator RICD" reveals the electronic structure of this solvation shell. So far, these two variants of RICD have only been identified theoretically in one system. The experimental observation of these RICD processes in a relevant system (liquid water) and the demonstration of their benefits for gaining information about solvation shells in liquids is valuable. The manuscript is well written, and the figures are clear. However, before we can consider recommending this manuscript for publication the following serious concerns must be dispelled.

Ca atoms and CaCl_2 have been studied by x-ray photoemission spectroscopy for a long time, see e.g. PRA 49 3685 (1994), PRB 47 11736 (1993).

1. The on-resonance electron spectra of Ca atoms (Fig. 7 of PRA 49 3685 (1994)) and CaCl_2 films (Fig. 5 in PRB 47 11736 (1993)) contain a feature between the Ca 3s and 3p photolines that resembles the feature assigned to "Normal ICD" or "Spectator ICD" in the current manuscript. What's the basis for the authors' claim that the feature they observe in the kinetic energy range 303-315 eV (Fig. 3) is due to ICD?
2. This same feature does not appear to be enhanced on resonance compared to the non-resonant case (Fig. 2 b, "normal ICD"). Only the edge of that feature around 312 eV kinetic energy is enhanced for high CaCl_2 concentrations and for $\text{Ca}(\text{NO}_3)_2$ solution. What's the basis of the authors' claim that this signal is due to "spectator RICD"? Is this concentration-dependent signal around 312 eV present off resonance (normal ICD) as well? If the authors want to uphold the claim that RICD is advantageous they have to clearly show/discuss this.
3. The authors should rationalize how the normal ICD / RICD signal of the Cl^- counterion can reach a level of $\sim 50\%$ of the H_2O ICD signal given the overall small concentration of Cl^- ions relative to water molecules (no depletion of the H_2O ICD signal is seen). Is ICD to Cl^- so much more efficient? Are there more than 2 Cl^- ions in the solvation shell around Ca^{2+} ? Is there any other signature of Counter-ion ICD that leads to detachment of the bound electron that forms the Cl^- anion?
4. The authors mention that electron inelastic scattering tends to be an issue when probing bulk liquid samples; What is the role of inelastic scattering of the participator RICD electrons, in particular in the discussion of the peak positions for participator ICD (Fig. 4)?
5. It is surprising to see a strong resonant enhancement of participator RICD of H_2O but no resonant enhancement of Cl^- (Fig. 4). Showing data for higher concentrations where participator RICD of Cl^- is also resonantly enhanced would make the case more convincing.
6. Overall, the discussion and conclusions obtained from these measurements remain purely qualitative. In the case of spectator ICD the only conclusion is that one previous observation is supported vs. another. The manuscript would certainly be more impactful if more quantitative information could be gained.

Reviewer #4 (Remarks to the Author):

Dear Editor,

We acknowledge receipt of the reviewer reports for our manuscript NCOMMS-24-07656-T. We thank the reviewers for their thorough work and their useful comments. We address these comments in the answers below. In addition, we made a number of modifications in the main text of the article, **highlighted in bold and blue** in the marked-up revised manuscript and in the answers below. We also now accompany the main article with a Supplementary Information file containing a substantial new amount of information based on the answers made below, additional spectra, and details on the analysis procedures.

Rémi Dupuy, on behalf of all authors.

Response to reviewers:

Reviewers #1 and #4 :

This paper shows how resonant intermolecular Coulomb decay (ICD) in liquid microjets has the potential to yield information about the electronic and geometric structure of the first solvation shell of ions and molecules in liquids. This is an important topic because the 1st solvation shell underpins many properties of, and processes in, solutions. It is therefore of interest to a broad chemistry audience.

The authors present a series of careful measurements of resonant ICD in Ca²⁺ solutions and describe clearly how the participator and spectator ICD processes have the potential to yield electronic and structural information, respectively, about the first solvation shell.

The paper is very well written so is accessible to a general chemistry audience. However, there are some points that could be clarified for it to be understood without recourse to the X-ray photoelectron spectroscopy literature.

We thank the reviewers for their positive assessment of our manuscript.

For example, Fig. 1 is very valuable in explaining the processes involved in this work. It would be even more useful if the caption could explain that H₂O / X⁻ in the second figure of Step 2 means that energy can apparently be transferred to either water or X⁻ in the spectator ICD, but not in the participator ICD. This was only stated explicitly on p5 of the manuscript.

It is also important that the authors explain why spectator ICD is a better probe of ion-pairing than participator ICD. This may be obvious to them, but to a non-expert in X-ray PES, it is not entirely obvious, and for this work to be published in a more general chemistry journal it would be nice to have it explained in the manuscript.

Both comments are connected and therefore we address both here. In fact, energy transfer to water or to Cl⁻ is possible for both spectator and participator ICD. However, transfer to Cl⁻ is only possible if ion pairs are formed (i.e., if Cl⁻ is sufficiently close to Ca²⁺). What we argue on page 5 is that ion pairing is absent in the case of 1.5 M CaCl₂ solutions, as previously shown using spectator ICD, and therefore the Cl⁻ photoelectron signal can be taken as a reference for the case of the absence of participator ICD. While this is true for this specific case, in general participator ICD with the counter-ion is possible, and thus participator ICD could also be used to probe ion pairing. However, it is much less convenient than spectator ICD because the participator signal always coincides with the regular photoemission signal,

while in spectator ICD it is manifested as a new feature well-separated from photoemission and other ICD/Auger signals.

To avoid confusion, we modified Fig. 1 so that it makes clear that participator ICD can also, in principle, involve the counter-ion. We clarified the discussion of this matter in the main text on page 5 (1st paragraph of the participator ICD section). Additionally, in the conclusion we now discuss the interest of spectator vs. participator (vs. other flavors) ICD for the detection of ion pairing in greater detail.

The discussion of the results is very clear. However, there are a couple of small things that should be addressed.

In the discussion of the effect of concentration (Fig. 3), the authors suggest that there is no ICD from the counter-ion in the 1.5 M spectrum, but it is not obvious from the data presented that this is necessarily the case. Was a spectrum recorded at weaker concentrations that had the same profile as the 1.5 M spectrum? If so, could this be reported in the text or the figure caption? If not, perhaps it would be better to state that there is less ICD from the counter-ion at 1.5 M rather than none.

We recently performed new experiments where we did measure lower concentrations. The spectra are reproduced below for the reviewers:

The suggested sentence was added to the manuscript, and this figure was added to the supplementary information (section S2.3). Note that this new data was taken using a different beamline and setup with (most importantly) a different X-ray polarization, and we thus do not attempt to make a full comparison with the data in the manuscript. The comparison of the 1.4 M and 0.7 M CaCl₂ data shown above and in the new Fig. S5 of the SI supports the statement that there is less (or even no) ICD from the counter ion in these data.

Since the point is made about the peak positions and widths being able to reveal information about the electronic and geometric structure of the solvent shell, it would have been useful to have some quantitative idea of how reliable the widths are that are presented in Table 1. The authors note twice in the manuscript how they are very sensitive to the subtraction process—so some more reassurance about this would be valuable. Moreover, if the widths are very sensitive to the subtraction process, aren't the peak positions and relative intensities of the various peaks also sensitive to the subtraction process? Perhaps the sensitivity study of all these parameters, which has presumably been undertaken already, should be presented as Supplementary Information to support the arguments in the manuscript.

We agree with the reviewers' suggestion that the sensitivity of the parameters and the uncertainties should be better explained. **We have added a section in the SI** which details the different sources of uncertainties for the values given in Table 1, and which also explains the dependence of the peak widths on the subtraction procedure. We redirect the reviewers to section S1 for more details. **We also updated the caption of Table 1, and redirected the reader to the SI for more details on this topic in the participator ICD section of the text (page 6 3rd paragraph and page 7 last paragraph).**

In Table 1, it is stated that the error on the measurements is 0.03 eV; however, the water 3a₁ peak positions from the measurements reported in this paper (row 2) are not within the error limits of the literature values (row 1). Perhaps the authors could comment on this or revise their error bars.

This discrepancy most likely arises from taking the 3a₁ peak positions from earlier works (e.g., Seidel et al. (2016), Ref. 42 in the manuscript), where the valence spectrum of water contained residual gas-phase components, contrary to more recent works where gas-phase contributions are completely smeared out by applying a bias voltage. For better consistency, we have compared our data only to the results of Thürmer et al. (2021), Ref. 43 in the main manuscript, which does not explicitly provide the 3a₁ peak positions, but they were retrieved from the publicly available raw data. We find a better agreement of the 3a₁ positions. **Table 1 was updated accordingly.**

Finally, in the Methods, the authors state that most measurements are made with a bias voltage of 50 V. What is meant by 'most'? And have the authors considered whether the bias voltage of 50 V has an impact on the alignment of the water molecules and counter-ions around the Ca²⁺ ions being studied?

We apologize for the confusion regarding the applied bias. Some of the spectra during the measurement campaigns were measured with a -60 V bias voltage instead of the -50 V bias voltage which was used in most cases, but all spectra shown in the manuscript were actually taken with a -50 V bias voltage, so **we corrected the sentence.**

Regarding the effect of applying a bias voltage to the microjet: from the point of view of the ions inside the liquid this should not have any particular effect as the electric potential remains uniform within the jet. Bias voltages are discussed e.g. in Stermer et al. 2023 (doi: 10.1063/5.0155182).

Reviewers #2 and #3:

This manuscript 'The solvation shell probed by resonant intermolecular Coulombic decay' by Dupuy reports an experimental study on the first solvation shell of Ca²⁺ solvated in CaCl₂ aqueous solution using liquid-jet photoemission spectroscopy. To selectively probe the solvation shell of water molecules surrounding the Ca²⁺ divalent ion, the authors study two variants of resonant ICD (RICD) following core level 2p_{3/2} resonant excitation of Ca²⁺ in the aqueous solution. They claim that "spectator RICD" informs about the chemical composition of the first solvation shell (also the signature of ion pairing between Ca²⁺ and its anion partner) and that "participator RICD" reveals the electronic structure of this solvation shell. So far, these two variants of RICD have only been identified theoretically in one system. The experimental observation of these RICD processes in a relevant system (liquid water) and the demonstration of their benefits for gaining information about solvation shells in liquids is valuable. The manuscript is well written, and the figures are clear.

We thank the reviewers for their positive assessment of our work.

However, before we can consider recommending this manuscript for publication the following serious concerns must be dispelled.

Ca atoms and CaCl₂ have been studied by x-ray photoemission spectroscopy for a long time, see e.g.

PRA 49 3685 (1994), PRB 47 11736 (1993).

1. The on-resonance electron spectra of Ca atoms (Fig. 7 of PRA 49 3685 (1994)) and CaCl₂ films (Fig. 5 in PRB 47 11736 (1993)) contain a feature between the Ca 3s and 3p photolines that resembles the feature assigned to “Normal ICD” or “Spectator ICD” in the current manuscript. What’s the basis for the authors’ claim that the feature they observe in the kinetic energy range 303-315 eV (Fig. 3) is due to ICD?

The reviewers rightly point out that our manuscript currently lacks a reference to previous work on atomic Ca and solid CaCl₂. We have therefore added the relevant references (refs. 40 and 41 of the revised manuscript).

Regarding the features observed between the Ca 3s and Ca 3p lines in both cases:

- For atomic Ca, these features were attributed to “L_{2,3}M_{2,3}N₁” Auger decay in Ref. 40 i.e., an Auger decay that involves the 4s electrons of Ca. Solvated Ca²⁺ ions have no 4s electrons, thus such a process is absent in our case.

- For the CaCl₂ films, the features are attributed to “LMV” Auger processes in Ref. 41, i.e., an Auger decay that involves the valence electrons of the system. Formally, Ca in solid CaCl₂ is also present in the form of Ca²⁺ ions devoid of 4s electrons, since CaCl₂ is an ionic solid, and therefore the valence electrons of the system are contributed by the Cl⁻ anions. One could argue that the “LMV Auger” process in this case is related to ICD, or at least it is an “interatomic Auger” process, similar to processes that were observed in ionic solids in the 1970s and 1980s (e.g. Citrin et al. 1976, doi: 10.1103/PhysRevB.14.2642) before the notion of ICD was formalized by Cederbaum et al. (1997, doi: 10.1103/PhysRevLett.79.4778). In aqueous solutions, ions are part of a weakly bound system, and thus the process by which water and/or Cl⁻ valence electrons participate in the decay of Ca²⁺ ions is more appropriately called ICD, although the features observed in both cases are indeed quite similar in shape and position.

As a side note, it is precisely because the Ca²⁺ ions in water have no outer valence electrons (the outermost orbital is the 3p orbital at ~30 eV binding energy) that it is possible to observe the local Auger and non-local ICD processes so well separated in the spectra.

We revised the “Resonant Auger and ICD spectra” subsection (page 4) where the assignments of the spectral features are discussed to reflect what we have developed here.

2. This same feature does not appear to be enhanced on resonance compared to the non-resonant case (Fig. 2 b, “normal ICD”). Only the edge of that feature around 312 eV kinetic energy is enhanced for high CaCl₂ concentrations and for Ca(NO₃)₂ solution. What’s the basis of the authors’ claim that this signal is due to “spectator RICD”? Is this concentration-dependent signal around 312 eV present off resonance (normal ICD) as well? If the authors want to uphold the claim that RICD is advantageous they have to clearly show/discuss this.

The fact that the spectator ICD feature does not appear to be enhanced compared to normal ICD derives from the fact that (almost) all other features in the spectra are resonantly enhanced as well since they (almost) all arise from Ca relaxation processes. Nonetheless, the resonant spectrum is recorded much more rapidly than the off-resonant spectrum (of the order of a few minutes vs. 1 hour for off-resonance) to obtain similar S/N ratios in the two spectra.

The resonant character becomes more evident when one looks at non-scaled spectra (or, alternatively, at the absorption spectrum in Fig. 2a). Below we display the on-resonance spectrum as well as the

spectra with excitation above and below the threshold. For better visibility the spectra are offset by the indicated amount, but not scaled along the counts/sweep axis.

The y scale in the graph above represents the raw counts per sweep (one sweep is one scan of the energy region on the detector, which lasts a few minutes). The above-threshold, off-resonance spectrum at 460 eV was recorded with a ten times higher photon flux compared to the other two spectra. The Auger count rate is therefore on the order of 200 times higher for the on-resonance spectrum. The same can be said of the ICD signal around 310 eV kinetic energy. We can therefore conclude that resonant (spectator) ICD is observed at 349.4 eV photon energy. **The figure shown above and this discussion were added to the SI (section S2.1); modifications mentioned above in the assignment section (page 4) also take into account this discussion.**

Is this concentration-dependent signal around 312 eV present off resonance (normal ICD) as well? If the authors want to uphold the claim that RICD is advantageous they have to clearly show/discuss this.

The concentration dependence due to ion pairing can also be observed in the off-resonance signal. We show the relevant figure below. This is, however, (i) less obvious for reasons we detail below and (ii) requires much more time to obtain sufficiently well-resolved spectra, as discussed above for the off-resonance above-threshold spectra. Such a well-resolved measurement was only possible at beamline P04 at synchrotron PETRA III because of the exceptionally high count rates that can be obtained from the combination of an efficient electron detection setup and the very high photon flux of the beamline; resonant spectra on the other hand can be readily measured using any of the liquid-microjet instruments installed at various other synchrotrons.

In addition to the longer acquisition time required to measure these spectra, additional complications can arise. Since the (off-resonant) ICD signal is weak, it is easily obscured by other signals such as photoelectron lines with similar kinetic energy. These can be tuned off by varying the photon energy to some extent, but data taken at too different photon energies become difficult to compare. Even the inelastic-scattering tail of these photoelectron lines, which extends often several tens of eV to lower kinetic energy below the line itself, can obscure the signal. For exactly this reason we could not properly measure the off-resonant ICD for CaI_2 and $\text{Ca}(\text{NO}_3)_2$ at 460 eV photon energy, due to interference of, respectively, the I 4p/4s lines and the N 1s Auger features. Another complication specific to the Ca^{2+} case is that there is already a feature at ~ 312 eV extending up to ~ 320 eV kinetic energy which does not correspond to normal ICD (either with water or the counter-anion). This feature, not present in the resonant ICD spectrum, comes from a satellite state (we plan to discuss this in future work). It further obscures the proper detection of counter-ion ICD.

All of this discussion was also added to the SI (section S2.2) to support the point that resonant spectator ICD is superior to normal ICD for the detection of ion pairing, and we also added reference to this discussion in the main text (page 5, end of the spectator ICD subsection, and also in the Conclusion).

3. The authors should rationalize how the normal ICD / RICD signal of the Cl^- counterion can reach a level of $\sim 50\%$ of the H_2O ICD signal given the overall small concentration of Cl^- ions relative to water molecules (no depletion of the H_2O ICD signal is seen). Is ICD to Cl^- so much more efficient? Are there more than 2 Cl^- ions in the solvation shell around Ca^{2+} ? Is there any other signature of Counter-ion ICD that leads to detachment of the bound electron that forms the Cl^- anion?

A quick analysis of the spectra of Fig. 3 (by subtraction of the 1.5 M CaCl_2 spectrum from the 5 M spectrum to isolate the Cl^- contribution) suggests that the RICD signal of Cl^- is on the order of $<20\%$ of the total RICD signal. We do not know the expected number ratio of Cl^- to H_2O in the immediate vicinity of Ca^{2+} , although it should be most likely between 1:4 and 1:8, therefore not so far from the ratio of RICD signals. It is nonetheless possible that ICD with the Cl^- ions is more efficient than with water. As discussed in the part dedicated to participator ICD, the ICD efficiency depends strongly on the interatomic distance and a favorable orbital orientation. While the Ca—Cl distance does not seem to be shorter than the Ca—O distance in concentrated solutions [Megyes et al., J. Phys. Chem. A **108**, 7261, (2004)], we can postulate that the larger extent of the Cl^- 3p orbital induces a more efficient process.

4. *The authors mention that electron inelastic scattering tends to be an issue when probing bulk liquid samples; What is the role of inelastic scattering of the participator RICD electrons, in particular in the discussion of the peak positions for participator ICD (Fig. 4)?*

There are two separate kinds of issues we (briefly) discussed. Relevant to our case of electron kinetic energies in excess of 100 eV is the detection of weak features in spectra that feature numerous peaks in close proximity: each strong peak gives rise to an inelastic-scattering background of electrons which starts out ~ 7 eV below the peak, with a maximum intensity at about ~ 20 eV below the peak and a gradually waning background beyond this energy. Weak features within the energy range of such inelastic-scattering tails are difficult to measure reliably. **This connects to the discussion now added in the SI about the detection of ion pairing with spectator vs. normal ICD (section S2.2, see also answer to point 2 above).**

For participator ICD, our data are not affected by this effect. Participator ICD occurs in the valence-band region, i.e., in the region of the highest-energy electrons of the spectrum, so no inelastic tail will occur in this region. At such high kinetic energy, vibrational inelastic scattering, i.e., very small energy losses (< 0.5 eV) that could lead to an asymmetric broadening of the spectrum, as has been observed for low-energy electrons (Malerz et al. 2021, doi: 10.1039/D1CP00430A), is entirely negligible because the vibrational inelastic-scattering mean free path is much larger than the electronic inelastic mean free path.

5. *It is surprising to see a strong resonant enhancement of participator RICD of H₂O but no resonant enhancement of Cl⁻ (Fig. 4). Showing data for higher concentrations where participator RICD of Cl⁻ is also resonantly enhanced would make the case more convincing.*

We did measure participator ICD data at high concentrations. Their analysis is complicated by the fact that we no longer have the possibility to normalize the spectra at the Cl 3p peak (as some participator ICD contribution to this peak is now expected) and we have instead to trust the absolute electron counts, which is subject to possible instabilities (in jet alignment for instance). In an example below where we believe we had stable conditions for spectra recorded back-to-back, we do observe an on-resonant enhancement of Cl⁻ (with a blue-shifted contribution, as expected). However, we did not use these data in the current manuscript because we believe more efforts are required to obtain reliable spectra at these high concentrations.

An interesting point which we had not noticed so far, is that the Cl⁻ participator ICD signal only represents $\sim 10\%$ of the total signal here, whereas the spectator ICD contribution of Cl⁻ is closer to 20% of the total signal. Such a discrepancy would need to be carefully confirmed and then investigated theoretically.

6. Overall, the discussion and conclusions obtained from these measurements remain purely qualitative. In the case of spectator ICD the only conclusion is that one previous observation is supported vs. another. The manuscript would certainly be more impactful if more quantitative information could be extracted.

The “spectator ICD / ion pairing” part of our work is certainly qualitative thus far, because many factors go into the intensities of the ICD signals and are not easily disentangled. On the other hand, the “participator ICD” part does provide a quantitative assessment of the electronic structure of solvation-shell water molecules around the Ca^{2+} ions. The core finding of our manuscript is the first experimental observation of the difference of the valence-band spectrum of water in the bulk versus water bound in a solvation shell. This information is vitally important for the benchmarking of theoretical models of water molecules in solvation shells and is presented here for the first time. Future experiments based on the methodological concept presented here will provide also more quantitative information. We also envision future theoretical investigations stimulated by our present (pioneering) experimental work, which will probably lead to more quantitative conclusions, for both spectator and participator ICD.

Editorial comments & other changes:

- Ref. 50 of the previous text (now Ref. 52) was updated as the corresponding article is now published.
- Very minor stylistic changes have been made to the text (choice of wording, typos).
- Data availability: we added the data availability section; the data is being prepared for upload to a Zenodo repository. The relevant DOI will be added to the section when this is done.
- We completed the required checklists and provide them in the resubmission files.

REVIEWERS' COMMENTS

Reviewer #2 (Remarks to the Author):

The authors have made a great effort in rebutting the critique of both referees; They added new experimental data and a Supplementary Information file. The manuscript is now ready for publication. Optionally, the authors may consider rephrasing the sentence "We can therefore also assume the absence of ICD in the participator ICD signal and normalize it to the Cl⁻ peak." as it sounds like a contradiction.